Genome-wide identification and characterization of the sucrose invertase gene family in Hemerocallis citrina

Ma Guangying mgyflower@163.com
Zuo Ziwei
Xie Lupeng
Han Jiao
Zhejiang Institute of Landscape Plants and Flowers , Hangzhou, Zhejiang , China
Domingues Douglas
Electronic publication date: 2024 Aug 29
Publication date: 2024
Volume: 12
Electronic Location ID: e17999
Received 2024 May 17; Accepted 2024 Aug 7
Copyright: © 2024 Ma et al.
Copyright year: 2024
Copyright holder: Ma et al.
License: This is an open access article distributed under the terms of the Creative Commons Attribution License, which permits unrestricted use, distribution, reproduction and adaptation in any medium and for any purpose provided that it is properly attributed. For attribution, the original author(s), title, publication source (PeerJ) and either DOI or URL of the article must be cited.
License URL: https://creativecommons.org/licenses/by/4.0/

Keywords: Daylily, Sucrose invertase, Genome-wide, Expression pattern

Funding: Zhejiang Provincial Natural Science Foundation LY20C160007 Hangzhou Xiaoshan District Science and Technology Bureau Project 2023103 This work was supported by the Zhejiang Provincial Natural Science Foundation (No. LY20C160007) and the Hangzhou Xiaoshan District Science and Technology Bureau Project (2023103). The funders had no role in study design, data collection and analysis, decision to publish, or preparation of the manuscript.

==============================
Background

Sucrose invertase is an important catalytic enzyme that is widely distributed in plants and can irreversibly hydrolyze sucrose into fructose and glucose. Daylily is an important perennial flower worldwide and a traditional vegetable in East Asia. Previous studies have suggested that sucrose invertase is involved in the aging of daylily flowers. However, knowledge about the number, physicochemical properties, and expression patterns of daylily sucrose invertases is still lacking. Identifying the daylily sucrose invertase family genes in the genome is highly important for understanding phylogenetic evolution and determining the genetic function of sucrose invertase.

Methods

To obtain basic knowledge about the number, classification, sequence composition, and physicochemical properties of sucrose invertases in daylily, bioinformatics software was used to analyze the genome of Hemerocallis citrina (H. citrina), and the basic properties of sucrose invertase genes and proteins were obtained. Then, combined with transcriptome data from flower organs at different developmental stages, the expression patterns of each gene were clarified. Finally, the reliability of the transcriptome data was verified by quantitative real-time polymerase chain reaction (PCR).

Results

Through software analysis, 35 sucrose invertases were identified from the H. citrina genome and named HcINV1-HcINV35; these enzymes belong to three subfamilies: cell wall invertases, vacuolar invertases, and chloroplast invertases. The amino acid composition, motif types, promoter composition, gene structure, protein physicochemical properties, gene chromosomal localization, and evolutionary adaptability of daylily invertases were determined; these results provided a comprehensive understanding of daylily invertases. The transcriptome expression profile combined with fluorescence quantitative reverse transcription-polymerase chain reaction (RT‒PCR) analysis suggested that almost all daylily invertase genes were expressed in flower organs, but even genes belonging to the same subfamily did not exhibit the same expression pattern at different developmental stages, suggesting that there may be redundancy or dissimilation in the function of daylily sucrose invertases.

Introduction

In higher plants, sucrose invertase can irreversibly hydrolyze the photosynthetic product sucrose into fructose and glucose, which are used for plant needs such as nutrition, energy, signal molecules, and stress response, which can increase plant yield (Mehdi et al., 2024). Based on the differences in the location of sucrose invertase in the cell, they can be roughly divided into cell wall invertases, vacuolar invertases, and cytosolic invertases. The optimal environmental conditions for cell wall invertases and vacuolar invertases are often acidic; hence, these enzymes are also known as acidic invertases. In contrast, cytoplasmic invertases are often found under neutral or alkaline conditions and are therefore also known as neutral/alkaline invertases (Wan et al., 2018). There have been many in-depth studies on acid invertases. Research has suggested that acid invertases play an important role in plant yield, especially in the formation of non-photosynthetic organs (sinks), cellular expansion, sugar accumulation, and stress response. However, research on neutral/alkaline invertases has focused mainly on its molecular evolution (Tauzin & Giardina, 2014; Wan et al., 2023). For invertase function, a cell wall invertase in tomato (CWIN) inhibited programmed cell death under long-term moderate heat stress through a non-ROS-dependent mechanism, thereby promoting sucrose importation in young fruits and helping maintain fruit set (Liu, Offler & Ruan, 2016). The transcription level and enzyme activity of another invertase gene in cucumber, CsVI2, are strongly induced by drought stress, and overexpression of this gene enhanced invertase activity and increased the drought stress tolerance of transformed plants (Chen et al., 2021). The acid invertase gene IbINV in sweet potato responded strongly to stress signals such as black rot disease and salicylic acid. Overexpressing this gene significantly improved plant growth and increased resistance to black rot disease, while downregulating its expression led to reduced plant growth and high sensitivity to black rot disease (Yang et al., 2023).

In addition to participating in the stress response, sucrose invertase also plays an important role in plant growth and development. Knocking down the expression of two Arabidopsis cell wall invertase genes (CWIN2 and CWIN4) using artificial microRNA resulted in arrested ovule development, while the expression of a group of hexose transporters was also downregulated in parallel with the two invertase genes; however, this regulation did not result in C starvation (Liao et al., 2020). In tomato, NI6 is a cytosolic neutral/alkaline invertase that is highly expressed in all organs, especially sink tissues. Silencing this gene significantly impaired plant growth, flowering, and fruit set. Multiomics analysis revealed that this gene plays a key role in sugar metabolism and stress adaptation (Coluccio Leskow et al., 2021).

In particular, sucrose invertase plays a critical role in the process of flower formation (Bolouri Moghaddam & Van den Ende, 2013) but also plays a key role in the opening of a single flower (Rabot et al., 2012). RhVI1, a glycosylated membrane-anchored protein derived from roses, is highly expressed in petals, and its expression level gradually decreases from buds to pre-senescent flowers (Farci et al., 2016). In dahlias, the activity of acid invertase was very low during the tight bud stage, sharply increased during the half-open stage of flowering, and then decreased during the fully open stage. Exogenous 6-BA can maintain the high activity of acid invertase, thereby delaying petal senescence to some extent (Shimizu-Yumoto, Tsujimoto & Terufumi, 2020).

Daylily (Hemerocallis spp.) belongs to the family Asphodelaceae and is native to China, East Asia, and other regions. Many species of Hemerocallis are traditional vegetables and medicinal herbs in China and some parts of East Asia, and the mature flower buds of daylily can be used directly for cooking after being picked or dried and stored for later consumption (Tian et al., 2017).

As a kind of traditional Chinese medicine, modern medical research suggests that daylily has great development potential in anti-depression (Xu et al., 2016), anti-oxidantion (Ti et al., 2022; Sang, Fu & Song, 2023), anti-inflammation (Hsu et al., 2023), diabetes treatment (Ti et al., 2022) and other health care functions. In addition to being used as a specialty vegetable and medicinal herb, daylily is used worldwide as an ornamental plant. Basic research on ornamental traits mainly focuses on aspects such as flower color (Li et al., 2021), flowering time (Liu et al., 2021; Ren et al., 2019), and stress response (Cai et al., 2023). Overall, daylily flowers, whether used for food, medicine, or ornamental purposes, have a lifespan of only one day, which poses significant challenges in terms of harvesting, storage, sales, and ornamental purposes. Proteomic studies have preliminarily confirmed that the expression of sucrose invertase changes dramatically during the development of daylily flowers as they open and close (Ma et al., 2018). However, to date, information on the sucrose invertase family in daylily remains incomplete, which poses a challenge for further elucidating the important role of invertase in maintaining flower longevity. This study was based on published genomic data for H. citrina (Qing et al., 2021) and involved mining and identifying the sucrose invertase gene family. Through a series of physicochemical property and expression analyses, this study provides a foundation for elucidating the mechanism by which invertases regulate flower longevity in daylilies.

Materials and Methods

Identification and physicochemical property analysis of the invertase gene family in H. citrina

First, we searched for protein sequences of the invertase family in Arabidopsis and rice (Ji et al., 2005; Sturm, 1999) using the NCBI online database. HMMER 3.0 software (Finn, Clements & Eddy, 2011) was subsequently used to construct a hidden Markov model using known INV protein family sequences, and this model was used to search for all protein sequences of H. citrina and identify all potential INV protein sequences. Subsequently, another software, BLAST (version: ncbi last v2.10.1+) (Altschul et al., 1990), was used to retrieve all sequences similar to those of Arabidopsis and rice invertases in the H. citrina database, with an e-value of 1e−5. The aligned sequences were all identified as potential invertase family sequences. Finally, the potential sequences from the above steps were merged to generate a candidate invertase family. All domains in the obtained sequences were annotated by the Pfamscan (version v1.6) and PfamA (version v33.1) databases (Finn et al., 2008, 2014) to define the final sequences, which should contain the PF12899, PF08244, PF00251, and PF11837 domains. Herein, 35 daylily invertase genes were identified and named HcINV1-HcINV35.

Phylogenetic analysis and multiple sequence alignment of the invertase family in H. citrina

The identified invertase protein sequences in daylily, Arabidopsis, rice, and foxtail millet, which included monocotyledons and dicotyledons, were used to construct a phylogenetic tree (protein sequences can be found in Data S1). In summary, multiple sequence alignment was performed via MAFFT (version v7.427), after which a phylogenetic tree was constructed via the neighbor-joining method via MEGA (MEGA10) software (Kumar et al., 2008). The parameter settings were as follows: the Poisson model was the basic model, partial deletion was the missing data method, the cutoff value was 50%, and the bootstrap value used to test the credibility of phylogenetic tree branches was set to 1,000. The software iTOL v6 (https://itol.embl.de/) was used to annotate the phylogenetic tree.

To compare the similarities and differences among the 35 invertase sequences more clearly, Jalview software (Waterhouse et al., 2009) was used to perform multiple sequence alignment on the complete invertase sequences, with the parameters set to default values.

Gene structure and protein motif analysis of the daylily invertase family

Gene structure is closely related to gene transcription and translation. To determine the gene structure and protein motif of the daylily invertase family, Gene Structure Display Server 2.0 (GSDS 2.0; https://gsds.gao-lab.org/) and MEME (version v5.05) were used for both analyses. Usually, there were numerous motifs in protein sequences; here, based on the display limitations and the importance of shared motifs, the top 15 motifs with the highest frequency were included in further analysis. The software operating parameters were set to the default settings. To better compare the gene structure with the motif composition of the corresponding protein, the two analysis results are displayed together.

Prediction and analysis of gene cis-acting elements, protein subcellular localization, transmembrane domains, and signal peptides

The 2-kb region upstream of the gene was selected as the promoter regulatory sequence, and the transcription factor-binding sites on the promoter were predicted using Plantcare (http://bioinformatics.psb.ugent.be/webtools/plantcare/html/). To highlight the main cis-acting elements, only the top 12 types of cis-acting elements were marked and displayed in the physical map of the gene promoters. The possible protein subcellular localization was predicted using WoLF PSORT (https://wolfpsort.hgc.jp/). The software DeepTMHMM (version 1.0.8) (Hallgren et al., 2022) is a program that predicts transmembrane helices based on deep learning models, integrating properties such as hydrophobicity, charge bias, helix length, and topological constraints of membrane proteins to analyze where a protein has a transmembrane domain. SignalP (version v5.0b) (Almagro Armenteros et al., 2019) is a signal peptide prediction software that is based on multiple artificial neural network algorithms and was used to predict whether there was a potential signal peptide region in the protein structure. The above software operating parameters were set to defaults.

Chromosome location, collinearity analysis, and evolutionary pressure analysis of the daylily invertase gene family

Based on the identified sequence information, MG2C (http://mg2c.iask.in/mg2c_v2.1/) was used to map the physical location of each gene on the chromosome, with the default parameters set. Gene family collinearity analysis was performed using MCSCANX (Wang et al., 2012) with the following main parameter settings: match_score 50; match_size 5; gap_penalty −1; overlap_window 5; e_value 1e−05; and max gaps 25. Herein, we selected representative monocots and dicots (rice and Arabidopsis, respectively) as controls to analyze the interspecies collinearity of the invertase gene family in daylily. Finally, the evolutionary pressure analysis of HcINVs was performed by caculating the nonsynonymous (Ka) to synonymous (Ks) substitution ratio of duplicated genes in H.citrina using the KaKs Calculator (version 3.0) software (Zhang, 2022).

Transcriptome expression profile and fluorescence quantitative RT–PCR verification of daylily invertase gene expression in flower organs

The plant material was 3-year-old clump seedlings planted at the research base of the Zhejiang Institute of Landscape Plants and Flowers. During the peak flowering period in June 2023, the flower buds of daylily were divided into four developmental stages: the juvenile stage (S1), the pre-opening stage (S2, at 8:00 on the day of full opening), the fully opened stage (S3, at 20:00 on the day of opening), and the post opening stage (S4, at 8:00 on the day after full opening) (Fig. S1). Three independent biological replicates of the material were taken during each stage and were frozen in liquid nitrogen and immediately ground for RT‒PCR and library construction. Except for sample preparation, the standardized operational procedures involved in transcriptome sequencing could be referred to Wan et al. (2024), and the relevant data were stored in the NCBI database with BioProject ID: PRJNA1050801. The total of 12 samples subjected to transcriptome sequencing resulted in 109.85 Gb of high-quality clean data. The clean data of each sample reached 6.8 Gb or higher, with Q20 base percentage above 96%, Q30 base percentage above 89%, and GC content between 43.9% and 45.8%. The proportion of reads generated by sequencing successfully aligned to the genome was higher than 79% (total mapped). To clarify the expression levels of each daylily invertase gene, fragments per kilobase of transcript per million fragments mapped (FPKM) values were used as indicators to measure the expression levels of the transcripts. In this process, the featurecounts program (Liao, Smyth & Shi, 2014) was used for FPKM calculations. The expression profiles of each gene in different developmental stages in daylily flowers are presented in the form of a cluster heatmap.

To verify the expression of the identified daylily invertase genes in flower organs at different stages, two genes were randomly selected from each subfamily of invertases and subjected to RT‒PCR analysis using the same sample preparation methods described above. The Total Plant RNA Extraction Kit (Tiangen, Beijing, China) was used for RNA extraction, and the extraction method was carried out strictly according to the manufacturer’s instructions. A Prime Script RT Kit (Takara, Dalian, China) was used and strictly implemented for cDNA synthesis, and the cDNA quality was detected by NanoDrop one spectrophotometer (Thermo Fisher Scientific, Waltham, MA, USA); the cDNA served as template for subsequent reactions after being diluted uniformly to a concentration of 100 ng·μl−1. Primer design was performed using Primer 3 combined with Primer Blast (NCBI) with the length of the amplification product between 80–200 bp, and was produced by Sangon Biotech (Shanghai) Co., Ltd. China. The sequences of primers can be found in Table S1, and the concentration of primers used was 1 μmol·L−1. The instrument used for fluorescence quantitative RT‒PCR was a Step One PlusTM Real Time PCR Instrument Thermal Cycling Block (Thermo Fisher Scientific Biotechnology, Shanghai, China). The reaction reagent kit used was SYBR Premium Ex TaqTMII (Takara, Dalian, China), with a recommended reaction system of 25 μl and a reaction program of 95 °C for 30 s, followed by 95 °C for 5 s and 60 °C for 30 s, for a total of 40 cycles. Data statistics and organization were completed in Excel (version 2010), and the relative expression levels of genes were determined by the 2 − ΔΔ CT method, using the expression of juvenile flower buds (S1) as a control. SPSS Statistics 19 was used to conduct statistical testing. Briefly, one-way analysis of variance (ANOVA) was conducted, with LSD and Tukey selected for post hoc tests, and the significance level set at 0.05. The differences between samples were indicated by lowercase letters, and different letters represented significant differences.

Results

Identification of the invertase gene family in H. citrina

A total of 35 invertase genes were identified as the final sequences from the genome of H. citrina through strict screening, using the invertase protein sequences from Arabidopsis and Oryza as reference sequences. These identified sequences were numbered HcINV1-HcINV35, and the gene ID, transcript ID, protein ID and chromosomal positions are shown in Table 1. The results showed that all 35 invertase genes had introns, with the highest number being 8 and the lowest being 1. The basic physicochemical properties of each invertase protein were determined through software analysis. Among the 35 invertases, HcINV26 had the greatest number of amino acid residues, while HcINV24 had the least, with 668 and 260 residues, respectively. The protein instability indices of 15 invertases, HcINV1, HcINV2, HcINV9, HcINV12, HcINV13, HcINV17, HcINV19, HcINV21, HcINV23, HcINV24, HcINV27, HcINV28, HcINV34, and HcINV35, were greater than 40, indicating poor stability; the remaining members had good stability. All the aliphatic indices of the invertases were less than 100, indicating that these proteins were hydrophilic. The corresponding grand average hydropathicity of each protein was negative, further indicating the hydrophilicity of the proteins. More detailed data, including the CDS length, molecular weight, and isoelectric point of each family member, can be found in Table 2.

Table 1 Identified genes in the daylily invertase gene family.

Gene	gene_id	Transcript_id	Protein_id	strand	Position on chromosome	Exon_num	Intron_num	
HcINV1	HHc003072	HHC003072.4	HHC003072.4	+	Superscaffold1:165164390–165170097	5	4	
HcINV2	HHC003073	HHC003073.1	HHC003073.1	+	Superscaffold1:165308591–165315258	7	6	
HcINV3	HHC003075	HHC003075.5	HHC003075.5	+	Superscaffold1:165484460–165503973	6	5	
HcINV4	HHC003083	HHC003083.2	HHC003083.2	+	Superscaffold1:166021976–166030864	9	8	
HcINV5	HHC003084	HHC003084.5	HHC003084.5	+	Superscaffold1:166150195–166157512	9	8	
HcINV6	HHC008919	HHC008919.2	HHC008919.2	–	Superscaffold2:285447581–285463227	8	7	
HcINV7	HHC008923	HHC008923.2	HHC008923.2	–	Superscaffold2:285596047–285603554	7	6	
HcINV8	HHC008924	HHC008924.6	HHC008924.6	–	Superscaffold2:285772188–285778795	8	7	
HcINV9	HHC011785	HHC011785.2	HHC011785.2	–	Superscaffold3:1727641–1733132	6	5	
HcINV10	HHC011823	HHC011823.3	HHC011823.3	+	Superscaffold3:3631114–3639215	4	3	
HcINV11	HHC011827	HHC011827.1	HHC011827.1	+	Superscaffold3:3748703–3752487	5	4	
HcINV12	HHC011978	HHC011978.4	HHC011978.4	–	Superscaffold3:13578248–13587046	4	3	
HcINV13	HHC016018	HHC016018.2	HHC016018.2	+	Superscaffold3:324919313–324930217	5	4	
HcINV14	HHC016149	HHC016149.2	HHC016149.2	–	Superscaffold3:328822064–328827512	7	6	
HcINV15	HHC016150	HHC016150.1	HHC016150.1	–	Superscaffold3:328894601–328911443	4	3	
HcINV16	HHC016151	HHC016151.1	HHC016151.1	–	Superscaffold3:328921057–328924788	4	3	
HcINV17	HHC016152	HHC016152.1	HHC016152.1	–	Superscaffold3:328939938–328947154	8	7	
HcINV18	HHC016486	HHC016486.2	HHC016486.2	+	Superscaffold4:9633824–9648706	8	7	
HcINV19	HHC019746	HHC019746.1	HHC019746.1	+	Superscaffold4:190020647–190045958	4	3	
HcINV20	HHC021014	HHC021014.1	HHC021014.1	+	Superscaffold5:10932188–10944471	7	6	
HcINV21	HHC027218	HHC027218.1	HHC027218.1	+	Superscaffold6:189673726–189679842	4	3	
HcINV22	HHC028998	HHC028998.1	HHC028998.1	+	Superscaffold6:286988387–286995310	7	6	
HcINV23	HHC029121	HHC029121.1	HHC029121.1	–	Superscaffold6:290471102–290500801	6	5	
HcINV24	HHC032515	HHC032515.1	HHC032515.1	+	Superscaffold7:265427913–265437396	2	1	
HcINV25	HHC038903	HHC038903.1	HHC038903.1	+	Superscaffold9:167952982–167958378	7	6	
HcINV26	HHC038910	HHC038910.1	HHC038910.1	–	Superscaffold9:168241058–168245847	7	6	
HcINV27	HHC041677	HHC041677.2	HHC041677.2	+	Superscaffold10:132156099–132171557	6	5	
HcINV28	HHC045402	HHC045402.1	HHC045402.1	+	Superscaffold11:57087676–57099116	5	4	
HcINV29	HHC047028	HHC047028.2	HHC047028.2	+	Superscaffold11:209925659–209997979	7	6	
HcINV30	HHC047557	HHC047557.1	HHC047557.1	+	unanchor80:2016197–2023004	7	6	
HcINV31	HHC048796	HHC048796.1	HHC048796.1	+	unanchor645:597145–600292	6	5	
HcINV32	HHC048797	HHC048797.1	HHC048797.1	+	unanchor645:661911–670172	5	4	
HcINV33	HHC051930	HHC051930.1	HHC051930.1	–	unanchor318:71002–77731	7	6	
HcINV34	HHC053023	HHC053023.1	HHC053023.1	–	unanchor260:350811–358786	4	3	
HcINV35	HHC054127	HHC054127.1	HHC054127.1	+	unanchor621:35236–46727	5	4	

Table 2 Physicochemical properties of daylily invertase members.

Invertase name	Amino acid num	CDS length	Molecular weight (Da)	Isoelectric points	Instability
index	Aliphatic index	GRAVY	
HcINV1	474	1,425	52,767.2	5.7	41	76.94	−0.319	
HcINV2	647	1,944	71,392.03	5.27	40.18	78.96	−0.219	
HcINV3	651	1,956	72,473.55	5.74	38	82.06	−0.272	
HcINV4	637	1,914	71,572.78	5.39	38.44	82.2	−0.245	
HcINV5	281	846	31,945.06	5	31.4	85.62	−0.321	
HcINV6	573	1,722	64,257.37	4.88	37.28	87.82	−0.171	
HcINV7	549	1,650	61,523.22	5.06	32.68	87.56	−0.178	
HcINV8	640	1,923	71,703.67	5.13	41.66	81.19	−0.286	
HcINV9	639	1,920	71,892.37	6.49	47.67	79.59	−0.273	
HcINV10	526	1,581	59,206.67	5.25	38.04	79.11	−0.359	
HcINV11	480	1,443	54,384.69	7.24	34.77	76.12	−0.384	
HcINV12	543	1,632	61,718.9	6.01	44.26	85.67	−0.181	
HcINV13	575	1,728	65,287	5.91	46.02	87.03	−0.137	
HcINV14	526	1,581	59,198.86	6.51	34.19	74.89	−0.431	
HcINV15	566	1,701	63,373.12	5.32	38.64	76.63	−0.327	
HcINV16	559	1,680	63,230.16	5.66	38.41	76.05	−0.351	
HcINV17	611	1,836	69,572.61	5.1	41.45	81.98	−0.369	
HcINV18	635	1,908	70,518.79	5.41	32.59	87.34	−0.193	
HcINV19	435	1,308	49,440.65	7.62	49.86	84.6	−0.309	
HcINV20	617	1,854	68,700.7	5.39	37.36	86.56	−0.206	
HcINV21	555	1,668	63,248.92	6.53	48.51	83.66	−0.193	
HcINV22	578	1,737	65,371.34	8.76	31.07	77.73	−0.418	
HcINV23	576	1,731	64,987.98	6.1	60.18	82.62	−0.282	
HcINV24	260	783	30,012.38	6.25	59.24	79.62	−0.282	
HcINV25	654	1,965	71,673.2	5.01	37.99	76.77	−0.281	
HcINV26	668	2,007	73,849.82	5.18	39.55	79.97	−0.284	
HcINV27	592	1,779	67,203.29	7.12	46.89	91.94	−0.237	
HcINV28	449	1,350	51,410.74	5.94	59.76	79.09	−0.303	
HcINV29	597	1,794	67,100.22	8.27	30.11	79.18	−0.363	
HcINV30	578	1,737	65,351.4	8.91	30.36	78.08	−0.413	
HcINV31	571	1,716	64,625	6.4	37.24	78.49	−0.376	
HcINV32	506	1,521	57,122.84	7.21	28.95	79.78	−0.291	
HcINV33	576	1,731	64,265.5	5.46	30.68	84.44	−0.25	
HcINV34	543	1,632	61,732.92	6.01	44.42	85.86	−0.18	
HcINV35	449	1,350	51,452.82	5.94	59.55	79.73	−0.297	

Subclassification and phylogenetic analysis of the invertase family in H. citrina

The identified protein sequences of invertases in daylily were combined with the invertase sequences of Arabidopsis, Oryza and Setaria to construct a phylogenetic tree and clarify the classification. The phylogenetic tree based on sequence composition was highly consistent with the functional localization results of the sequences. Briefly, the 35 invertases identified in this study were divided into five branches based on evolutionary distance (Fig. 1). Among these branches, the vacuolar invertase (Va) branch had the greatest number of members (13). The remaining nine members belonged to the chloroplast-II (Chl-II) branch, two members belonged to the Chl-I branch, six members belonged to the cell wall-III (CW-III) branch, and five members belonged to the CW-II branch; only the invertase members from Arabidopsis belonged to the CW-I branch, while rice, foxtail millet, and daylily did not have this type of invertase. Similarly, the CW-II branch contained only monocotyledonous members of rice, foxtail millet, and daylily but did not contain any invertase members from Arabidopsis. The phylogenetic tree well illustrated the relationships between monocotyledons and dicotyledons and provides a reference for further research on the functional divergence of invertases in daylily.

Figure 1 Phylogenetic relationship of the invertase family genes in daylily.

The subfamilies of vacuolar invertase, chloroplast invertase, and cell wall invertase are abbreviated as Va, Chl, CW, respectively. The red triangle represents Arabidopsis thaliana. The green circle represents Setaria italica. The yellow square represents Oryza sativa.

To more intuitively examine the sequence similarity and structural conservation of the 35 daylily invertases, multiple sequence alignment was performed for all members. The results revealed significant differences in sequence length and structural composition among the different daylily invertases. Taking HcINV1 as the benchmark, HcINV2 had the highest similarity, with a score of 93.038%, while the lowest similarity was found for HcINV19, with a similarity of 10.114% (Data S2). According to the alignment results, the amino acid composition of each invertase significantly changed at the N- and C-termini, while the middle region was relatively conserved, and multiple sequence alignments showed high similarity (Fig. S2). Protein composition similarity analysis clearly demonstrated specific differences between different invertases and provided a reference for understanding functional divergence and sequence evolution.

Protein motif and gene structure analysis of the daylily invertase family

Gene structure and motif composition are highly important for studying the evolution, functional divergence, and subfamily attribution of transcription factors. Figure 2 shows the gene structure and protein motif composition of 35 daylily invertases. In this figure, the gene structure horizontally corresponded to the motif composition. The results showed that all 35 invertase genes contained varying numbers of introns, and two genes, HcINV4 and HcINV5, contained the largest number of introns, with eight introns each, while HcINV24 contained only one intron. Multiple introns in one gene provided the possibility for gene splicing to produce different mature mRNAs. The results also revealed that some genes contained superlong introns, such as HcINV23 and HcINV19, which resulted in gene lengths exceeding 24 kb, and inevitably, both of these genes belonged to the chloroplast-localized subfamily.

Figure 2 Diagram of the gene structure and protein motifs of daylily invertase family genes.

For protein motif analysis, MEME software was used to identify the motifs of all daylily invertases, and the results showed that among the 15 motifs with the most widespread distribution and the greatest number of motifs, motif 1 appeared in 24 invertases, which are vacuolar invertases or cell wall invertases. Interestingly, all neutral/alkaline invertases (Chl-I and Chl-II) did not contain motif 1. Motif 10 and motif 14 were present in 10 invertases, all of which belonged to the neutral/alkaline invertase family. Furthermore, of the 11 identified neutral/alkaline invertases, only HcINV24 did not contain motif 10 or motif 14. The distribution of motifs was closely related to the evolution and clustering of the invertase family, and specifically distributed motifs could serve as an important reference for the subclassification of invertases. The motif compositions of 35 inverted genes were largely consistent with the gene clustering results, indicating that genes with similar exonic and intronic compositions had highly similar motif compositions. The specific quantity of motifs and amino acid composition of each invertase are shown in Data S3. The logos of all 15 motifs are shown in Fig. S3.

Prediction and analysis of cis-acting elements in the daylily invertase family

The cis-acting elements distributed on the promoter played a key role in the transcriptional regulation of genes. Plantcare software was used to identify and extract the promoter regions of the 35 daylily invertase genes, and a total of 90 cis-acting elements were identified under the set conditions. The results indicated that some cis-acting elements had multiple copies in the same promoter, leading to significant differences in element quantity. The most abundant cis-acting element was the CAAT-box element, with 1,048 copies in total, while the least abundant had only one copy (Data S4). Among these cis-acting elements, at least 25 were related to light response, which was the largest category. Among the hormone response elements, one was related to abscisic acid (ABRE), two were related to salicylic acid (TCA-element and SARE elements), four were related to auxin (AuxRR-core, AuxRE, TGA-box, and TGA-element), three were related to gibberellin (TATC-box, p-box, and GARE-motif), and two were related to jasmonic acid (CGTCA-motif and TGACG-motif). In addition, there was one cis-acting element related to circadian rhythm (circadian). The top 12 cis-acting elements with the highest abundance were selected and further statistically analyzed, and a heatmap showing the distribution of each element in gene promoters is shown in Fig. 3. Among the 12 cis-acting elements that occurred with the highest frequency, the cis-acting elements with clear and specific functions were mainly light response elements and hormone response elements. There were two cis-acting elements that were functionally specific and related to the light signal response: one was the G-box element, and the other was the box-four element, which appeared 145 times and 83 times, respectively. The former did not appear in only two invertase promoters (HcINV2 and HcINV29), while the latter did not appear in five promoters (HcINV6, HcINV19, HcINV22, HcINV27, and HcINV30). There was a total of 176 ABREs related to the abscisic acid response; similarly, only two genes, HcINV2 and HcINV29, did not contain this element. The CGTCA and TGACG elements related to the jasmonic acid response were each found 62 times, with 10 of each in the HcINV6 gene promoter, where the occurrence frequency was the highest. The detailed distribution of the 12 cis-acting elements on the promoters is shown in Fig. 4. Overall, from the perspective of both the distribution and number of cis-acting elements, the light response and hormone response were the most important signals in regulating the expression of daylily invertase.

Figure 3 Quantitative statistics of the 12 cis-acting elements in each gene promoter.

Figure 4 Distribution sites of 12 cis-acting elements on the promoter.

Prediction and analysis of the subcellular localization, transmembrane domain and signal peptide of the invertase family in H. citrina

The subcellular localization, transmembrane domain, and signal peptide are highly important for predicting and understanding the physicochemical properties and functions of daylily invertases. In this study, the above contents were predicted and analyzed using corresponding software. In terms of subcellular localization, the results showed that daylily invertases could be localized to three different cellular organs to play a specific role, namely, the cell wall, vacuole, and chloroplast. According to the prediction results, there were 16 invertases localized in the cell wall, 11 invertases localized in the chloroplast, and eight invertases localized in the vacuole. No invertase was localized in other subcellular structures, and the prediction results were consistent with the phylogenetic tree results. Proteins can be divided into soluble proteins and membrane proteins based on whether they contain transmembrane domains. According to software analysis, of the 35 daylily invertases, only HcINV1, HcINV2, HcINV4, HcINV8, HcINV18, HcINV25, and HcINV26 contained distinct transmembrane domains and were membrane proteins. SignalP was used to identify signal peptides based on probability values. Generally, a probability value >0.5 was considered to indicate the presence of a signal peptide. The prediction results showed that among the 35 daylily invertases, the signal peptide probabilities were highest for HcINV15, HcINV16, HcINV20, HcINV22, HcINV30, HcINV31, and HcINV32, and the results also showed that invertases localized in chloroplasts generally lacked signal peptides, while invertases with a high probability of containing signal peptides were cell wall invertases. The above analysis results can be found in Table 3.

Table 3 Analysis of the subcellular localization, transmembrane domain, and signal peptide of daylily invertases.

Alpha TM, including the transmembrane helix. Globular, all amino acids inside or outside the membrane. Globular+sp, all amino acids with signal peptides inside or outside the membrane.

Invertrase ID	Localization	Transmembrane type	Signal peptide (probability)	
HcINV1	Vacuole	Alpha TM	0.043854	
HcINV2	Vacuole	Alpha TM	0.004966	
HcINV3	Vacuole	Globular	0.001179	
HcINV4	Cell wall	Alpha TM	0.004278	
HcINV5	Cell wall	Globular	0.006056	
HcINV6	Cell wall	Globular	0.08105	
HcINV7	Cell wall	Globular+sp	0.100682	
HcINV8	Cell wall	Alpha TM	0.001995	
HcINV9	Chloroplast	Globular	0.001232	
HcINV10	Cell wall	Globular	0.002324	
HcINV11	Cell wall	Globular+sp	0.41037	
HcINV12	Chloroplast	Globular	0.000989	
HcINV13	Chloroplast	Globular	0.000793	
HcINV14	Cell wall	Globular	0.020756	
HcINV15	Cell wall	Globular+sp	0.573799	
HcINV16	Cell wall	Globular+sp	0.583438	
HcINV17	Cell wall	Globular	0.001208	
HcINV18	Vacuole	Alpha TM	0.011302	
HcINV19	Chloroplast	Globular	0.000901	
HcINV20	Vacuole	Globular+sp	0.894949	
HcINV21	Chloroplast	Globular	0.001164	
HcINV22	Cell wall	Globular+sp	0.985938	
HcINV23	Chloroplast	Globular	0.483258	
HcINV24	Chloroplast	Globular	0.004784	
HcINV25	Vacuole	Alpha TM	0.007433	
HcINV26	Vacuole	Alpha TM	0.00084	
HcINV27	Chloroplast	Globular	0.000423	
HcINV28	Chloroplast	Globular	0.000818	
HcINV29	Cell wall	Globular+sp	0.171003	
HcINV30	Cell wall	Globular+sp	0.986979	
HcINV31	Cell wall	Globular+sp	0.878318	
HcINV32	Cell wall	Globular+sp	0.944255	
HcINV33	Vacuole	Globular+sp	0.009601	
HcINV34	Chloroplast	Globular	0.000989	
HcINV35	Chloroplast	Globular	0.000818	

Chromosome location, collinearity analysis, and evolutionary adaptability of daylily invertase genes

Utilizing published genomic data for H. citrina, the chromosomal distribution and location of the daylily invertase gene family were confirmed (Fig. 5). The results showed that 29 of the 35 daylily invertase genes were dispersed on 10 different chromosomes, and the remaining six invertase genes were located on ‘unanchored scaffolds’. The genes on the chromosome are arranged in the following order: Superscaffold3 (nine genes, accounting for 25.7%), Superscaffold1 (five genes, accounting for 14.3%), Superscaffold2 (three genes, accounting for 8.6%), Superscaffold6 (three genes, accounting for 8.6%), Superscaffold4 (two genes, accounting for 5.7%), Superscaffold9 (two genes, accounting for 5.7%), Superscaffold11 (two genes, accounting for 5.7%), Superscaffold5 (one gene, accounting for 2.9%), Superscaffold7 (one gene, accounting for 2.9%), and Superscaffold10 (one gene, accounting for 2.9%); six HcINV were genes located on “unanchored scaffolds”, accounting for 17.1% of these genes.

Figure 5 Physical location map of daylily invertase genes on chromosomes.

From the distribution of HcINV genes on the chromosomes, it could be seen that there were multiple HcINV gene clusters on the chromosomes of Hemerocallis citrina (Fig. 5). In these gene clusters, a total of four pairs of tandem duplicate genes were identified, based on MCScanX collinearity analysis, namely HcINV1-HcINV2, HcINV7-HcINV8, HcINV15-HcINV16, HcINV16-HcINV17. Further analysis of intraspecies collinearity revealed 29 pairs of genes with segment duplication, visualized through Circos, the results were shown in the Fig. 6. Among these 29 pairs of collinear genes, six genes (HcINV10/11/12/13/14/15) were concentrated on chromosome 3, with the higest number. Additionally, HcINV22 and HcINV30 each formed five pairs of segment duplicate genes (HcINV22-HcINV11/14/29/30/32; HcINV30-HcINV11/14/22/29/32), with the most frequent gene duplications. Hence it could be seen that segment duplication rather than tandem duplication was the main type in the duplication process of HcINV gene family.

Figure 6 Visualization of intraspecific collinearity of the daylily invertase gene family.

A interspecies collinearity analysis was conducted on daylily and model plants Arabidopsis and rice. The results showed that there were three genes in daylily, HcINV14, HcINV22, HcINV30, which had collinearity with three homologous genes in Arabidopsis and rice, respectively. The visualization was shown in the Fig. 7 (Due to the genome quality, HcINV30 was unable to accurately locate, therefore not shown in Fig. 7). Analysis of the Ka/Ks ratio between members of the daylily invertase gene family revealed that all the ratios were less than 1 (Data S5), indicating that the number of sites undergoing synonymous substitution during the evolution of daylily invertase genes was greater than the number of sites undergoing nonsynonymous substitution (Ka>>Ks) and that the genes were subjected to purifying selection.

Figure 7 Synteny analyses of daylily invertase genes between Arabidopsis thaliana and Oryza sativa.

The syntenic invertase gene pairs between daylily and other species are highlighted with red lines.

Gene expression patterns at different flower developmental stages

To clarify the expression patterns of the invertase gene family in different flower developmental stages in daylily, the corresponding early transcriptome data were extracted, and a gene expression heatmap was drawn according to the FPKM values. As shown in Fig. 8, almost all 35 daylily invertase genes were ubiquitously expressed in flower organs, but the expression levels of different genes at different developmental stages were distinct. Moreover, even if some invertases were located in the same subcellular structure, their expression patterns still significantly differed. In other words, daylily invertase genes that were located in the same subcellular structure might be highly expressed at the early flower developmental stage, middle flower developmental stage, or flower decay stage. The peak expression of daylily invertase genes may occur at any of the four flower developmental stages, almost certainly indicating that invertases play critical roles in the development of flower organs from infancy to senescence. In addition, the cluster heatmap of gene expression patterns revealed that there was often more than one gene with the same expression pattern, suggesting that gene function redundancy might be common in daylily invertases.

Figure 8 The expression profile of invertase genes in different flower development stages in daylily.

Based on the different subcellular localizations of daylily invertases, two cell wall invertase, vacuolar invertase, and chloroplast invertase genes were selected for subsequent RT‒PCR validation. Among the selected genes, HcINV3 and HcINV6 are vacuolar invertase genes, HcINV11 and HcINV15 are cell wall invertase genes, and HcINV9 and HcINV34 are chloroplast invertase genes. Then, the reliability of the transcription profile was verified using fluorescence semiquantitative RT‒PCR. The validation results (Fig. 9) showed that the expression trends of all genes were basically consistent with the transcriptome data, indicating the reliability of both the transcriptome and RT‒PCR data. Moreover, the results confirmed that genes with the same subcellular localization may exhibit inconsistent expression and function.

Figure 9 Gene expression validation by real-time RT‒PCR.

Discussion

Sugar, as an important photosynthetic product, is synthesized in the green organs of plants and then distributed to various sink organs (such as roots and reproductive organs) as the main energy source for life activities, used for a series of growth and development activities (Braun, 2022). Sucrose is the most important form of carbon assimilation in green plants. Once sucrose is distributed into the sink cells, it will be quickly broken down into monosaccharides, such as fructose and glucose, or converted into relatively inert storage compounds, such as starch (Salvi et al., 2022; Zhu et al., 2024b).The enzymes involved in sucrose metabolism mainly include sucrose invertase and sucrose transporter. Nevertheless, the gene families related to sugar metabolism have been less studied and studies have mainly focused on Sugars Will Eventually be Exported Transporter (SWEET), Sucrose Transporter (SUT), and a few other gene families (Hu et al., 2021; Wu et al., 2022; Zhang et al., 2022; Sun et al., 2023). Sucrose invertases are important node enzymes in the plant sugar metabolism pathway and are also important response factors that plants use to respond to stress and modulate yield. Therefore, an increasing number of sucrose invertase families have been identified in several important industrial crops (Dahro et al., 2021; Wang et al., 2022; Abbas et al., 2022). However, there have been no studies on the identification and expression of invertase family members in the flower organs of Asphodelaceae and its relatives.

Here, a total of 35 members of the daylily invertase family were identified, which was significantly greater than that of other diploid species (Dahro et al., 2021; Zhang et al., 2021; Abbas et al., 2022). Previous studies have suggested that the number of invertase members, especially those in the cytoplasmic subfamily, increased significantly during the evolution of higher plants from lower plants. This increase in number is positively correlated with both plant growth rate and biomass and is also closely related to whole-genome duplication events (Wan et al., 2023). Daylilies are perennial herbaceous plants that have thick fleshy roots. More importantly, approximately 15.73 million years ago, they experienced a whole-genome duplication event, leading to a significant increase in the copy number of orthologous genes (Qing et al., 2021). The larger size of the genome and greater biomass are likely directly related to the greater number of invertases in daylilies. Previous bioinformatics statistics confirmed that nearly all known chloroplast invertases contain motif 3, motif 10 and motif 14, whose amino acid compositions are presented in this article; however, HcINV24 in daylily, which does not contain motif 10, was an exception. This finding was highly consistent with the 11 chloroplast invertases identified here, indirectly confirming the accuracy of the phylogenetic tree and gene family number in this study.

Compared to phylogenetic analysis of different gene sequences, collinearity analysis could better utilize genomic data and more effectively identify evolutionary relationships between species (Fu et al., 2024). Paralogous genes located on different chromosomal segments of a species are defined as segment duplications (Han et al., 2017). Gene clusters generated by fragments of gene divisions are generally considered tandem duplication (Wang et al., 2010). Tandem duplication is often an important source of new gene copies in gene clusters, while segment duplication can expand genes more widely (Lin et al., 2024). There were both gene tandem duplication and segment duplication in the daylily HcINVs, but segment duplication were predominant, indicating the coexistence of conservation and expansion in the evolution process of HcINVs within the species. Due to limited molecular research on closely related species of daylily, studies on collinearity between multiple species were limited. Interspecies collinearity analysis revealed only three HcINV genes in daylily, which were found to be collinear with homologous genes in Arabidopsis and rice, respectively. The small number reflected a diversification of sequence composition and function of the HcINV family during evolution.

Among the identified response elements, the light response elements dominated in the promoters of the daylily invertase genes, highlighting the important role of light response in the transcriptional regulation of daylily invertase. Flower organs have few chloroplasts, and their vital activities mainly rely on energy transfer from photosynthetic organs (such as leaves) because of their carbohydrate requirements. In fact, as part of the sink, invertase cleavage of sucrose predominates in flowers where carbohydrates are catabolized for respiration and has been associated with cell expansion (Winter & Huber, 2000). Moreover, the key influence of light signals on the expression and activity of invertase in different plant sink organs has been frequently reported (Rabot et al., 2014; Klopotek et al., 2016; Zhang et al., 2021). Therefore, light signals likely play important roles in regulating the expression of invertase genes in daylily flower organs, and the presence of invertases directly regulates fluctuations in sucrose, fructose, and glucose contents, which can directly affect the growth, opening, and longevity of flower organs.

There is evidence that sugars must interact with hormone signals to form complex metabolic networks that regulate sink organ activity (Shen et al., 2020). Here, the cis-acting elements of daylily invertase genes included a large body of plant hormone response motifs, second only to light response elements, which respond to auxins, gibberellins, salicylic acid, abscisic acid, and jasmonic acid signals; this result was consistent with existing knowledge. The known promoter regions of plant invertase genes contain numerous hormone response elements that can regulate the expression of invertase genes and ultimately play key roles in plant growth and development, stress responses, and other activities required for life (Xu et al., 2022; Mao et al., 2024; Zhu et al., 2024a). Moreover, plant hormones, such as IAA, cytokinins, and ABA, have been shown to act as signaling molecules that regulate sugar (such as starch) accumulation, either individually or in concert, thereby affecting the sink activity of reproductive organs (Du et al., 2023); furthermore, a decrease in the expression level or activity of invertases directly leads to a decrease in the sink activity of reproductive organs (Sosso et al., 2015). This evidence suggested that there must be a close relationship between invertase expression and hormones and that hormones likely regulate the sink activity of reproductive organs by binding to cis-acting elements of invertase genes.

The plant sucrose invertase genes underwent significant changes in sequence composition due to selection pressure during long-term evolution, as reflected by the large variation in the number of introns and exons. Previous studies have shown that there are 23 sucrose invertase genes in pear trees, with the number of exons ranging from four to eight the acidic invertase genes contain five to eight exons and the neutral invertase genes contain four or six exons (Wu et al., 2020). The number of identified exons in maize invertase genes ranged from three to nine (Juárez-Colunga et al., 2018), while the number in Dendrobium huoshanense ranged from four to seven (Song et al., 2023). The number of exons identified in the daylily invertase genes in this study ranged from two to nine (Table 1), reflecting significant differences in the number of exons among different species. Even within relatively conserved subfamilies, characteristic motifs were easily lost during evolution. For example, acid invertases generally contain a short exon consisting of nine bases that encodes a conserved DPN motif, but deletions occur in acid invertases from maize and poplar (Chen et al., 2015). In this study, nine of 24 acid invertases contained this conserved motif, but another conserved MWECPD motif was found in all members. During evolution, functional differences may have acted as selection pressures, leading to significant changes in the structural composition of sucrose invertases in different species.

Sucrose invertase genes constitute a large gene family with significant functional differences, reflected not only in the large quantitative differences among different species but also in the vast differences in expression and function among members of the same family. Previous studies on the expression pattern of sucrose invertase genes have focused on their changes in response to biotic and abiotic stresses (Zhu et al., 2024a; Mao et al., 2024; Li et al., 2024); among the 126 sucrose invertase genes identified in wheat, 70 responded to stress signals (Wang et al., 2022). However, recent studies have shown that sucrose invertase is commonly expressed in plant roots, stems, leaves, four whorl floral organs, and even young fruits and plays key roles in plant growth and development. Among the 12 sucrose invertase genes identified in cucumbers, several are strongly expressed in reproductive organs, suggesting that they are closely related to organ development (Qi et al., 2023). Further research confirmed that expression of the CsCWIN3 gene in cucumber dramatically changes during flower organ development and is essential for various stages of pollen development, fertilization, and fruit development in cucumber (Liu et al., 2024). Here, almost all the daylily invertase genes were expressed in flower organs, but the expression patterns of each gene differed, and even the expression patterns of members belonging to the same subfamily were not the same. This situation was similar to that of invertase genes in other crops, which are ubiquitous but have distinct functions. For day-opening and night-wilting daylily flowers, the period before and after opening (S2–S3) was the critical stage for the formation and maintenance of quality flowers. At this stage, a total of four vacuolar invertase genes (HcINV1, −2, −25, and −26), two cell wall type III invertase genes (HcINV29 and −32), and one chloroplast invertase gene (HcINV9) showed significant changes in expression, while their expression levels were generally low at other developmental stages, indicating that high expression of these genes was beneficial for the opening and maintenance of flowers. Whether these genes directly control daylily flower opening and lifespan through sugar metabolism should be further studied.

Conclusions

Sucrose invertase plays an important role in plant source‒sink metabolic processes. To better understand the role of daylily invertase throughout the process of flower organ development, 35 H. citrina invertase genes were identified at the whole-genome level, and were classified into three subfamilies. The knowledge about physicochemical properties, structural composition, chromosome localization, and gene evolution of these genes were clarified through software analysis. The gene expression analysis suggested that daylily invertase genes are ubiquitously expressed in flower organs, but different genes and subfamilies have different expression patterns. This work lays a foundation for further study of the functional mechanisms of daylily invertases in flower organ growth and development.

Supplemental Information

Supplemental Information 1 Schematic diagram of H. citrina flowers.

Supplemental Information 2 Multiple sequence alignment diagram of 35 daylily invertases.

Supplemental Information 3 The 15 motif logo sequences used in this study.

The motif code is provided above the y axis.

Supplemental Information 4 The 15 motif logo sequences used in this study.

The motif code is provided above the y axis.

Supplemental Information 5 The 15 motif logo sequences used in this study.

The motif code is provided above the y axis.

Supplemental Information 6 Sequences of primers used in this study.

Supplemental Information 7 The amino acid sequences of invertase in daylily, Arabidopsis, rice, and foxtail millet.

Supplemental Information 8 Comparison of amino acid sequence similarity of 35 daylily invertases.

Supplemental Information 9 The motifs found in 35 daylily invertases.

Supplemental Information 10 Analysis of cis-elements within the promoters of invertases gene family members.

Supplemental Information 11 Ka/Ks analysis for daylily invertase gene pairs.

Supplemental Information 12 MIQE checklist.

Supplemental Information 13 RT-PCR raw data.

Supplemental Information 14 Melt curve for target genes.

We are immensely grateful to the editors for their comments on the manuscript.

Additional Information and Declarations

Competing Interests

Author Contributions

Data Availability

The authors declare that they have no competing interests.

Guangying Ma conceived and designed the experiments, performed the experiments, analyzed the data, prepared figures and/or tables, authored or reviewed drafts of the article, and approved the final draft.

Ziwei Zuo performed the experiments, prepared figures and/or tables, authored or reviewed drafts of the article, and approved the final draft.

Lupeng Xie performed the experiments, authored or reviewed drafts of the article, and approved the final draft.

Jiao Han performed the experiments, analyzed the data, authored or reviewed drafts of the article, and approved the final draft.

The following information was supplied regarding data availability:

The raw sequence reads are available at NCBI: PRJNA1050801.

The raw PCR data are available in the Supplemental File.

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
