# Peer review of "Genome-wide identification and characterization of the sucrose invertase gene family in Hemerocallis citrina"

_PeerJ, doi:10.7717/peerj.17999_

## Round 0.1 · original submission · Major Revisions

· Academic Editor

Major Revisions

Reviewers stress that the discussion section needs significant improvement in the manuscript. If this improvement is not evident in future versions, the manuscript will not be further assessed. Authors must address and clarify all reviewer comments.

Reviewer 1 ·

Basic reporting

No comment

Experimental design

No comment

Validity of the findings

No Comment

Additional comments

The article submitted by Guangying Ma et al. provides a comprehensive study entitled "Genome-wide identification and characterization of the sucrose invertase gene family in Hemerocallis citrina". The paper is in detail wrote however it contains irrelevant information. The interpretation is markedly poor across the whole of the article's result and discussion parts. However, there are a few randomly selected issues stated below:

1. The manuscript lacks appropriate justification and fails to incorporate significant material. However, the paper contains excessive and unnecessary information, particularly in the introduction (lines 53 to 81) and the discussion (lines 427 to 438).
2. Please include the required citation from lines 83 to 93.
3. The interpretation of tandem and segmental duplication in lines 198 to 204 is ambiguous to me.
4. In line 216 to 220 the author generates transcriptome data (including Bio Project ID) to interpret invertase gene family. But can’t provide any information regarding to transcription analysis.
5. I feel it tough to comprehend lines 206 to 208.
6. In RT-PCR analysis the used unit is not correct, primer 3 is used for primer design not synthesis.
7. The naming of invertase genes must be based on their corresponding genes in Arabidopsis and Rice.
8. Line 231 Typo.
9. The author present the gene HcINV5 and HcINV5 has highest 8 introns according to the manuscript and table, however, in the figure 2 gene HcINV5 and HcINV5 shows 6 intron.
10. The author reported 21 duplication event out of a total of 35. I am uncertain about the accuracy of this information and would appreciate further clarification in the discussion section.
11. The conclusion is rather lengthy and should be more succinct.

·

Basic reporting

No comment

Experimental design

No comment

Validity of the findings

No comment

Additional comments

General comment
Title: Genome-wide identification and characterization of the sucrose invertase gene family in Hemerocallis citrina
Authors have done a wonderful work on the Genome-wide identification and characterization of the sucrose invertase gene family a very important medicinal plant, Daylily.
The research manuscript has followed the standard language and scientific writing format throughout the document.
It has identified 35 putative sucrose invertase sequences from daylily and characterized using different in silico tools. Phylogenetically compared with other model plants in monocot and dicot family. The chromosomal localizations, the collinearity analysis,gene structure, motif and cis-acting DNA binding element (CREs) in the promoter regions, and the prediction of the subcellular localizations. The authors further carried the expression analysis using fluorescence quantitative RT‒PCR for verification. This justifies that the experiment is properly designed and authors have done excellent work.
The in silico tools employed in the study for identification and characterization are well known elsewhere in the literature. And authors also applied some additional tools in their work for characterization. Further they have also validated their results using the quantitative RT-PCR.
This is a wonderful piece of work on daylily to improve the quality of the its product on the flower, and tips with interesting scholarly information regarding the geneome-wide characterization invertase gene family H. citrine.
The Manuscript is bear very well substantiated/narrated points on the topic, well designed in the methods, and validate the findings from the in-silico identification and characterization with RT-PCR.

Reviewer 3 ·

Basic reporting

The paper combines genomics and transcriptomics data of daylily to uncover the number, physicochemical properties, and expression patterns of daylily sucrose invertases. Overall the paper is interesting to the readers in plant science.

Some minor mistakes:
1. Please spell out the abbreviation such as RT-PCR, UHPLC-MS
2. The font size in line 212-224 is not consistent with other main text.
3. Please add figure legend to fig. 1. Which species and subfamily those color and shapes represent?

Experimental design

In line 159-161: Please explain what is the rationale to select such parameters.
In line 171: Please explain the criteria to determine 15. To highlight is not a sufficient reason.
Fig 9: Please conduct statistical testing for it.

Validity of the findings

The conclusions are well stated and linked to original research question. But please increase the image resolution.

---

## Round 0.2 · accepted · Accept

· Academic Editor

Accept

Authors have addressed all of the reviewers' comments.

Reviewer 1 ·

Basic reporting

No Comments

Experimental design

No Comments

Validity of the findings

No comments

Reviewer 3 ·

Basic reporting

The revised manuscript has clear and unambiguous professional English used throughout.

Experimental design

The revised manuscript has addressed my previous concerns on the hypothesis testing.

Validity of the findings

The revised manuscript has its conclusion well stated.